# Development of Cost-Effective, Ecofriendly Selenium Nanoparticle-Functionalized Cotton Fabric for Antimicrobial and Antibiofilm Activity

**Kainat Mirza [1], Farha Naaz [2], Tokeer Ahmad [2], Nikhat Manzoor [3] and Meryam Sardar [1,\*]**

[1] Enzyme Technology Lab, Department of Biosciences, Jamia Millia Islamia, New Delhi 110025, India
[2] Nanochemistry Laboratory, Department of Chemistry, Jamia Millia Islamia, New Delhi 110025, India
[3] Medical Mycology Lab, Department of Biosciences, Jamia Millia Islamia, New Delhi 110025, India
\* Correspondence: msardar@jmi.ac.in

**Abstract:** In the present study, selenium nanoparticles were synthesized in situ on alkali-activated cotton fabric using guava leaf extract as a reducing agent. The synthesis was monitored by a change in color of fabric from white to light brick red. The UV-DRS analysis confirms the coating of Se NPs on cotton. The XRD, FT-IR, and SEM-EDX characterization techniques were used to analyze the nanoparticles on cotton fabric. The peak at 788 cm$^{-1}$ in FT-IR confirms the formation of Se NPs on cotton fabric. The XRD analysis confirms that the average crystallite size of as-prepared nanoparticle is ~17 nm. SEM-EDX analysis shows the successful coating of Se NPs on coated fabric. ICP-OES studies confirm 3.65 mg/g of selenium nanoparticles were present on the fabric. The Se-coated-30 showed a larger zone of inhibition against Gram-positive *S. aureus* (32 mm) compared to Gram-negative strains of *E. coli* (16 mm) and *K. pneumoniae* (26 mm). The fabric was also tested against the fungi *C. glabrata* (45 mm), *C. tropicalis* (35 mm), and *C. albicans* (35 mm) and results indicate it is more effective against fungal compared to bacterial strains. The coated fabric inhibits biofilm formation of *C. albicans* (99%), *S. aureus* (78%), and *E. coli* (58%). The results demonstrated excellent antibacterial, antifungal, and antibiofilm activities of the Se-coated-30. The prepared fabric has the potential to be used in medicinal applications and is both ecofriendly and cost effective.

**Keywords:** cotton fabric; alkali; selenium nanoparticles; green synthesis; guava; antimicrobial; biofilm; durability

## 1. Introduction

Nanomaterials are a diverse array of materials with dimensions between 1 and 100 nm [1]. Due to the size in the nanometer scale, they have various properties that are often inaccessible with bulk or molecular materials. In biomedicine, nanoparticles are widely used in drug delivery, magnetic resonance imaging, bone growth promoters, cancer treatments, biocompatible coatings for implants, sunscreens, biolabeling, antiviral, antibacterial, and fungicides, for example [2–4]. The main disadvantage of using metal nanoparticles now is the high cost associated with their synthesis, which produces toxic by-products by traditional physico-chemical methods. Scientists have developed ecofriendly methods of manufacturing nanoparticles utilizing plant extracts, bacteria, and fungi to circumvent the effects of toxic chemicals. Plant extracts, for example, lower the cost of culture media and microbe isolation. In particular, phytochemicals, the backbone of plants, readily produce fewer toxic nanoparticles [5]. These phytochemicals prevent aggregation and prolong the stability of the nanoparticles. Hence, the method allows the synthesis of cost-effective and biocompatible nanoparticles compared to physical and chemical methods. Since India has a great diversity of medicinal herbs and shrubs, plant-mediated synthesis of nanoparticles can be adopted on a large scale [6]. In past studies plants have been explored

successfully for rapid biosynthesis of gold [7], silver [8], selenium [5], MgO [9], CuO [10], and ZnO nanoparticles [11].

Recently, Se NPs are emerging as a new member of nanomedicines because of their strong antioxidative, anticancer, and antibacterial activity [5]. There is some evidence that Se nanoparticles can selectively kill bacteria but leave mammalian cells intact. Se-based nanoparticles could more readily be accepted by a human than Ag- and Cu-based ones since Se is already present in humans as a trace element [12]. Previously, our lab synthesized colloidal selenium nanoparticles using *Psidium guajava* leaf extract. *Psidium guajava* contains many reducing and stabilizing agents needed for the green synthesis of Se NPs. Guava is an important fruit readily available in tropical countries such as India, Pakistan, Bangladesh, and Indonesia due to its diverse medicinal properties. Guava leaf extract is a rich source of Vitamin C/ascorbic acid and has numerous health benefits due to the abundance of beneficial phytochemical. Past studies have reported the various therapeutic applications of guava leaf extract such as anticancer, antioxidant, antidiabetic, antidiarrheal, antimicrobial, hepatoprotection, and lipid-lowering activities [13]. The green synthesized nanoparticles were confirmed by the color change from colorless to brick red; the synthesis took only 3 h and was non-toxic to human cell lines but effective against hepatic cancer cell lines [5]. This study showed a simple one-pot synthesis of nanoparticles using ecofriendly precursors of guava leaf extract and sodium selenite salt. The obtained nanoparticles were separated easily by centrifugation. Due to the simple protocol and ease of operation, inexpensive and green nanoparticles were formed.

With the application of nanotechnology to natural and synthetic materials, researchers are developing several types of nanomaterial finishes. Nano finishes have been used with cotton, wool, and silk along with polyethylene terephthalate (PET) fabrics to produce interesting results. Using nanostructured ZnO for UV-blocking [14], sodium montmorillonite clay as a flame retardant [15], silver for antimicrobial activity [16], silica for water repellency [17], TiO$_2$ for self-cleaning [18], silver and ZnO for conductivity [19,20], as well as protecting/modifying textile products, are some of the examples of their use. Nano-finished fabrics of this kind can reduce the cost, time, and chemicals while improving efficiency and versatility.

Past studies suggest textiles can act as carriers for microbes in healthcare settings [16]. Their large surfaces allow them to retain more moisture, and nutrients, making them ideal substrates for microbes [21]. The most widespread pathogens associated with healthcare settings are *Escherichia coli*, *Klebsiella pneumoniae, Staphylococcus aureus*, and *Candida* spp. [22,23]. The alarming increase in antibiotic resistance to conventional antimicrobial agents in the past few decades has become a common threat in both hospitals and community settings. This is especially applicable to biofilms, because bacteria encased in biofilm are more tolerant to antibiotics and drugs than sessile cells. Since biofilms are linked to bacterial adhesion, application of nano-finished surfaces such as zinc, silver, and gold has shown favorable results in lowering microbial adhesion followed by biofilm formation and, as a result, device-linked infection [24]. To this, Ag is the extensively used nanofinish on textiles due to its antimicrobial activity against broad range of microbes, but its acute cytotoxicity, low number of clinical trials, and growing concern of antibiotic resistance, limit its human usage. It is therefore important to discover and develop new antimicrobial nano finishes on textiles to protect the health of wearer [12,16].

During both in vitro and in vivo studies, Se NPs have been found to exhibit antimicrobial properties and have low cytotoxicity towards mammals. To date, not much work has been reported on the antifungal activity of Se NP-coated textiles. Yip et al. 2014 synthesized Se NP-functionalized fabrics using a polysaccharide–protein complex (PSP) derived from mushrooms as a stabilizer. The Se NPs were applied to fabrics through the pad-dry-cure method; these fabrics were significantly effective against *Trichophyton rubrum*. However, the antibacterial activity of Se NP-functionalized fabrics was not tested [25]. Biswas et al. 2018 synthesized silver- or selenium- loaded polymeric scaffolds and compared their cytotoxicity towards mouse fibroblasts cells through an indirect approach; the study showed

no cytotoxicity of Se scaffolds compared to that of Ag scaffolds [12]. Sadalage et al. 2020 synthesized selenium brooms (SeBrs) using almond skin extract and ascorbic acid and in situ coated them on cotton fabric; the fabric turned dark brown after coating. The coated fabric exhibits good antibacterial activity against *B. subtilis* [26]. In another study, selenium or silver nanoparticles were in situ coated on cationized cellulosic fabrics for antimicrobial action against *E. coli*, *S. aureus* and *K. pneumonia* [16]. The results of the study showed that Se -or Ag-coated fabrics exhibit good antibacterial activity and low cytotoxicity and demonstrated that Se NP-coated fabric is a good alternative to Ag NPs due to the reported high cytotoxicity of Ag. Elmaaty et al. 2022 synthesized Se NPs and dyed wool fabric using the immersion method; the Se NPs impart an orange color to the fabric. The authors reported good antimicrobial and UV-protection properties of the dyed fabric [27]. In a study in which Se NPs were decorated on polyester fabric through a screen-printing technique monitored via color change, the printed fabric exhibited strong antiviral activity against the SARS-CoV-2 virus as well as showed antibacterial activity with low cytotoxicity to human cell lines; however, the antifungal property of the decorated fabric was not studied [28].

In practice, the methods used to stabilize inorganic materials on textile surfaces require multistep preparation processes that are time-consuming and not economical. Sometimes the binding molecules needed for the adherence of the nanoparticles to the textile's surfaces are toxic; for instance, acrylic acid esters, epoxides, isocyanate groups, fluoropolymers, etc. [29]. In this light, it is essential to find new ways to create antimicrobial fabrics that are effective, cheap, and environmentally friendly.

In the present study, Se NPs were in situ synthesized on the surface of alkali-activated cotton fabric using sodium selenite salt solution and guava leaf extract. The method of the in situ synthesis of Se NPs is simple, cost-effective, and ecofriendly, as no harsh chemicals were used. The NPs were synthesized on the surface of a fabric in an environmentally friendly manner. The Se-coated fabric thus generated was assessed in terms of its washing durability, antibacterial, antifungal and antibiofilm performance.

## 2. Materials and Methods

### 2.1. Microorganisms and Culture Conditions

Bacterial strains *E. coli* (MTCC 405), *S. aureus* (MTCC 3160), and *K. pneumonia* (MTCC 10309) were obtained from Microbial Type Culture Collection (IMTECH, Chandigarh) and maintained in Luria broth at 37 °C and 180 rpm. Luria agar plates were used to maintain the culture at 4 °C. Fungal strains *C. albicans* (ATCC 90028), *C. glabrata* (ATCC 90030), and *C. tropicalis* (ATCC 750) were cultured on yeast extract peptone dextrose (YPD) medium containing 1% yeast extract, 2% peptone, and 2% (*w/v*) glucose (Hi-Media, India) at 37 °C and 200 rpm, and maintained in YPD agar plates containing 2.5% agar at 4 °C in the Medical Mycology Lab, Jamia Millia Islamia, New Delhi.

### 2.2. Chemicals

Sodium selenite ($Na_2SeO_3$) was obtained from Sigma Aldrich; sulfuric acid ($H_2SO_4$), potassium hydroxide (KOH), nitric acid ($HNO_3$), hydrogen peroxide ($H_2O_2$), and 99% absolute ethanol were of analytical grade. Luria Agar, Luria Broth, YPD media, and YPD agar were purchased from Hi-Media Laboratory.

Fresh guava leaves were plucked in the month of November from the university campus (Jamia Millia Islamia). Pure cotton fabric was obtained from the local market, New Delhi, India.

### 2.3. Coating of Se NPs on Cotton Fabric
#### Alkali Activation of Fabric

Before activation the fabric was washed with hot water to remove the impurities, then dried and kept in a desiccator. The fabric (1 cm × 1 cm) was activated with potassium hydroxide solution as described earlier [30,31]; for this fabric was kept in an alkali solution

(3 M) for 5 min and then rinsed thoroughly with distilled water. The activated fabric was coated with Se NPs in two ways.

(1)  Physical adsorption of colloidal Se NPs on cotton fabric;

A colloidal solution of Se NPs was made as described earlier in our lab [5]. The activated fabric was immersed in 1 mL of colloidal Se NPs under shaking at 120 rpm for 2 h at 25 °C, then fabric was taken out, rinsed thoroughly with distilled water and dried at 65 °C.

(2)  In situ synthesis of Se NPs on cotton fabric;

The activated fabric (1 cm × 1 cm) was immersed in 3 mL of different concentrations of sodium selenite salt solution (10–40 mM) separately and incubated at 50 °C under shaking at 140 rpm. After 1 h, all samples were taken out and washed thoroughly with distilled water to remove unbound salt. *Psidium guajava* leaf extract (10%) was added to each sample to reduce the selenium salt and kept for 3 h at 50 °C. The fabrics were taken out, washed thoroughly with distilled water to remove loosely bound Se NPs, and dried at 65 °C for 60–90 min. The obtained fabric was called Se-coated.

*2.4. Characterization of Se-Coated Fabric*

2.4.1. UV-Visible Diffuse Reflectance Spectroscopy (UV-Vis DRS)

A PerkinElmer Lambda 365 UV-Visible Spectrophotometer was employed to obtain reflectance spectra of the fabric samples in the wavelength range of 200–800 nm.

2.4.2. Fourier-Transform Infrared (FTIR)

The fabric samples were directly analyzed for their Fourier-transform infrared (FTIR) spectra, Cary 630 FTIR, Agilent Technologies (Denver, CO, USA).

2.4.3. X-ray Diffraction (XRD)

Fabric samples were directly loaded on to the sample holder and affixed with transparent tape on either side, leaving the center part, which was analyzed by monochromatized Cu Kα (k = 1.54051 A°) within the diffraction angle from 10° to 80° (2θ) radiation on a Rigaku Ultima IV X-ray diffractometer system (Tokyo, Japan). Average crystalline size was calculated by Scherrer's Equation (1) [32].

$$D = \frac{K\lambda}{\beta \cos \theta} \tag{1}$$

D: crystal size of Se NPs
B: full width at half maximum of the diffraction peak (Rad),
λ: wavelength of X-ray source
K: constant of Scherer equation (value range 0.9–1)
θ: used in Radian

2.4.4. Scanning Electron Microscopy (SEM) and Energy Dispersive X-ray Spectroscopy (EDS)

Surface morphologies of fabric samples were observed on a Zeiss SIGMA Field Emission Scanning Electron Microscope (FE-SEM) (Zeiss, Germany) equipped with an energy-dispersive X-ray spectroscope (EDS) at 15 kV. The surface chemistry of the samples was analyzed using EDS. For the EDX analysis, the samples were fixed with carbon tape and sputter-coated with a thin layer of gold to increase conductivity of samples.

2.4.5. Inductively Coupled Plasma-Optical Emission Spectrometry

Quantitative determination of selenium content on Se-coated samples was performed as described by Yan Bai et. al, 2015 [33]. Briefly, a 5 mL mixture of $m_{\text{HNO3}}/m_{\text{H2SO4}}/m_{\text{H2O2}}$, 20:10:3 was prepared in a beaker and a fixed size of Se-coated-30 (1 cm$^2$ = 0.023 − 0.025 g) was dissolved in the mixture. Then, the mixture was heated until the solution turned yellow

and a few drops of HNO$_3$ (3 mL) was added gradually till it turned colorless. The resulting solution was diluted, and the volume was made up to 50 mL with deionized water. The Se concentration (mg/L) was estimated by Perkin, Avio 200, ICP-OES. The amount of Se NPs loaded on the fabric surface was calculated using the Equation (2):

$$\text{Amount of Se NPs loaded (mg/g)} = \frac{C_{Se} \times V_{Se}}{W} \qquad (2)$$

where $C_{Se}$ is the concentration of *Se* (g/L) obtained after dilution, $V_{Se}$ (mL) is the volume obtained after dilution and *W* is the weight of cotton fabric (g).

*2.5. Biological Activity*

2.5.1. Determination of Antibacterial and Antifungal Property of Se-Coated Samples by Agar Plate Method

The antibacterial efficacy of UV-sterilized Se-coated samples was evaluated against *E. coli* (gram-negative), *S. aureus* (Gram-positive), and *K. pneumonia* (Gram-negative) by the JIS L 1902: 2008 method (testing method for antibacterial activity of textiles) [34]. Briefly, the fabric samples were UV-sterilized for 10 min and placed in close contact with Luria agar pre-seeded with an inoculum adjusted to 0.1 OD$_{600}$ of tested strains. The agar plate was incubated for 24 h at 37 °C and the zone of inhibition (ZOI) of fabric samples was measured diagonally along the fabric's surface. Similarly, the antifungal activity of UV-sterilized fabric samples was checked against *C. albicans*, *C. glabrata* and *C. tropicalis* by the agar plate method [35,36]. Simultaneously antimicrobial activity of control fabric samples, guava extract (10%) coated fabric, activated fabric (3 M KOH), salt-coated fabric (30 mM) and uncoated fabric, was also evaluated. All the experiments were performed in triplicate.

2.5.2. Determination of Antibiofilm Activity of Coated Fabric by Static Biofilm Method

The antibiofilm activity of the fabrics was determined as described earlier by Wang et al. 2020 [24]. Briefly, bacteria were cultured in Luria broth overnight. An OD$_{600}$ of 0.1 was kept constant for all bacterial culture's strains. For the biofilm assay, in a sterilized 24-well plate an equal amount of bacterial culture (1 mL) was added and then fabrics (1 × 1) cm$^2$ were added to these wells so that they completely submerged in the culture media and were incubated for 48 h at 37 °C; the media were changed with fresh culture media after 24 h. Next, the supernatant was discarded, and wells containing fabrics were rinsed twice with distilled water to remove the loosely bound cells. With the help of a cell scraper, the adhered cells were scraped out from the fabrics' surface in 250 µL of Luria broth. Finally, 200 µL of scraped cells from the 250 µL of Luria broth was taken out and transferred to a 96-well plate; the rest of the wells were filled with 180 µL of Luria broth to perform serial dilution. After serial dilution, 10 µL of aliquot was spread onto LB agar plates and incubated at 37 °C for 18 h and growth of bacteria was expressed in colony-forming units/mL (cfu/mL), as measured in a colony counter. Similarly, antibiofilm activity of fungal *C. albicans* was detected using YPD media and YPD agar. The growth reduction (R%) in the viable cell count on the fabric surface was also calculated using the Equation (3):

$$\text{R (\%)} = \frac{\text{UF} - \text{CF}}{\text{UF}} \times 100 \qquad (3)$$

where UF and CF are the cfu/mL of uncoated fabric and Se-coated fabric, respectively, after incubation.

2.5.3. SEM Analysis of Biofilm of Untreated and Se-Coated Fabrics

Untreated and Se-coated fabrics were incubated with *S. aureus* bacterial inoculum for 48 h as described above. After 48 h incubation, the samples were washed twice with autoclaved distilled water to remove loosely attached cells. The samples were fixed and dehydrated for SEM imaging as described earlier [16]. Briefly the untreated and Se-coated fabrics were fixed with 2.5% glutaraldehyde for 2 h at 25 °C. Additionally, they were

dehydrated in a series of graded ethanol solutions (35–100%) for 15 min each. The samples were then air dried for 2 days and examined in SEM as described in the Section 2.

### 2.6. Washing Durability

Fabric samples were tested for their durability to washing cycles according to the AATCC 61-2006 method with few modifications [16]. For the test, polypropylene wide-mouth bottles, 10 stainless steel balls (diameter 6 mm) and a washing detergent, Persil washing powder, were purchased from the local market, India. Polypropylene wide-mouth bottles were filled with 150 mL of distilled water, 10 stainless steel balls, along with Persil washing detergent (0.2% *w/v*). The bottles were heated to 40 °C and fabric samples (5 cm × 10 cm) were added to it. The bottles were shaken at 100 rpm for 45 min in an incubator shaker (one accelerated washing cycle) preset at 40 °C. According to AATCC Test Method 61-2006, this process of the accelerated washing cycle was equivalent to five home wash cycles. After 30 wash cycles, the fabric samples were thoroughly washed, dried naturally, and evaluated for their antibiofilm activity as described in the Section 2 above.

### 2.7. Statistical Analysis

The experiments were performed in triplicate and mean values of data are represented. Standard deviation is shown wherever required. The statistical significance among samples for multiple comparisons was tested by the Kruskal–Wallis test followed by the Mann–Whitney test for pairwise comparison. Additionally, the Student's *t*-test was performed to compare with untreated control wherever necessary. $p < 0.05$ is considered statistically significant.

## 3. Results and Discussion

### 3.1. Synthesis and Coating of Se NPs on Cotton Fabric

Previously, our lab reported synthesis of antimicrobial and non-toxic colloidal selenium nanoparticles using sodium selenite and *Psidium guajava* leaf extract [5]. Therefore, in the present study, Se NPs were coated on cotton fabrics to enhance the applications of these nanoparticles. It has been reported that pretreatment of cotton with alkali enhances the physical and chemical properties of cellulose fibers such as color strength, hydrophilicity, reactivity, and washing fastness [30,37]. Shateri et al. 2017 reported an increased deposition of silver nanoparticles on alkali-treated cotton fabric [38]. In a study by Nawaz et al. 2022, the surfaces of non-woven cotton fabrics were activated with KOH; they reported that activated fabric increased the deposition of Ag NPs and enhanced antibacterial activity compared to the non-activated one [31]. Thus, in this study, cotton was pretreated with KOH before being coated with Se NPs.

Se NPs were synthesized by incubating sodium selenite (25 mM) and guava leaf extract (5%), and kept for 3 h at 50 °C [5]. The resultant solution was centrifuged, washed with distilled water, and air-dried to obtain Se NPs. A colloidal solution of Se NPs was prepared by dispersing the Se NPs in distilled water. In the first experiment, activated cotton (1 cm × 1 cm) was incubated with 1 mL of colloidal solution of Se NPs for 2 h, and the cotton fabric turned red in color, indicating the adsorption of Se NPs on the cotton surface. When this sample was washed with distilled water, all the color and Se NPs leached out from the cotton, which shows that Se NPs were loosely adsorbed on the surface of cotton; hence, no further experiments were done with this sample.

In the second experiment, activated fabric was immersed separately first in different concentrations of sodium selenite salt solution (10–40 mM) followed by guava leaf extract for reduction of sodium salt. When 10% *Psidium guajava* leaf extract was added to these samples, they turned red within 3 h. The results indicate that at all the concentrations, the selenite gets reduced to nano selenium; this was assessed visually from the color change of the cotton fabric. The control cotton fabric remains colorless. The proposed mechanism of in situ synthesis of Se NPs on cotton fabric is depicted in Figure 1. Alkali activation causes the hydrolysis of H-bonds present in cellulosic cotton and converts them

to potassium cellulosate (Cell.O⁻K⁺) and Cell.OH. The fabrics were washed with water to remove excess unbound KOH. When sodium selenite was added, it reacted with free OH groups (Cell.O⁻Se). In a subsequent reaction, selenium ions were reduced with guava leaf extract, which causes the in situ synthesis of Se NPs on cellulosic cotton. A similar mechanism was proposed by Yazdanshenas et al. 2012 for the in situ synthesis of silver nanoparticles on alkali-activated cotton [30].

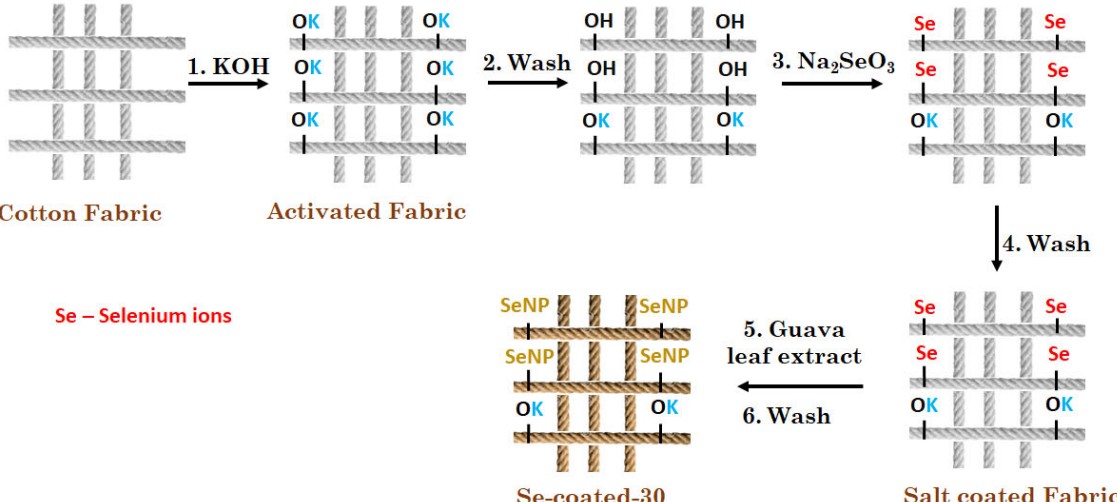

**Figure 1.** Illustration showing the steps involved in the in situ synthesis of Se NPs on cotton fabric. (1) Activation of fabric with KOH. (2) Washed with distilled water to expose free hydroxyl groups. (3) Treatment of the fabric with sodium selenite salt. (4) Washed to remove unbound salt. (5) Reduction of selenium ions by guava leaf extract.

*3.2. Characterization Studies*

3.2.1. UV-Visible Diffuse Reflectance Spectroscopy

Diffuse reflectance spectroscopy was performed to investigate the optical properties of uncoated and coated fabric samples in the range of 200 to 800 nm. To evaluate whether an applied coating of cotton can act as a UV-protective material, the investigation was done in the UV region of 220 to 400 nm. The visible region of 400 to 750 nm was investigated to analyze the changes in coloration of the as-prepared coated fabrics. The reflectance data provide the evidence of color changes after coating with nanoparticles, and an uncoated cotton fabric is employed as a reference. Figure 2 displays the reflectance spectra of the uncoated, alkali-activated, and Se NP-coated cotton fabrics with varied concentrations of Se NPs. The concentration of sodium selenite salt varied from 10 mM to 40 mM on the activated fabric, followed by a reduction with 10% guava leaf extract. The coated samples are referred to as Se-coated-10, Se-coated-20, Se-coated-30, and Se-coated-40, depending on the salt concentration used for synthesis. Because of the differences in sodium selenite salt concentration, reflectance spectra were compared. It was obvious that the Se coating decreased the reflection over the entire spectral range from 200 to 800 nm. Such a decrease in reflectance is evidently associated with the concentration of added selenium coatings. It is expected that the reflection in the visible spectrum of light will diminish due to the brown color of the Se-coated fabrics. Additionally, the brown coloring of the fabric is responsible for the higher reflection values for red light in comparison to lower reflection values for light in the 400–500 nm range. Dark-brown-colored samples result from coating applications with increasing sodium selenite concentrations. Along with this effect on visible light, the reflection of UV light is also significantly reduced. Samples with higher selenium concentrations show reflection values that are less than 5% almost over the whole UV spectrum. The applied coatings have a smaller impact on IR light reflection than they do on UV and visible light reflection. As a result, it may be stated that infrared light reflects light more strongly than the other types of light analyzed.

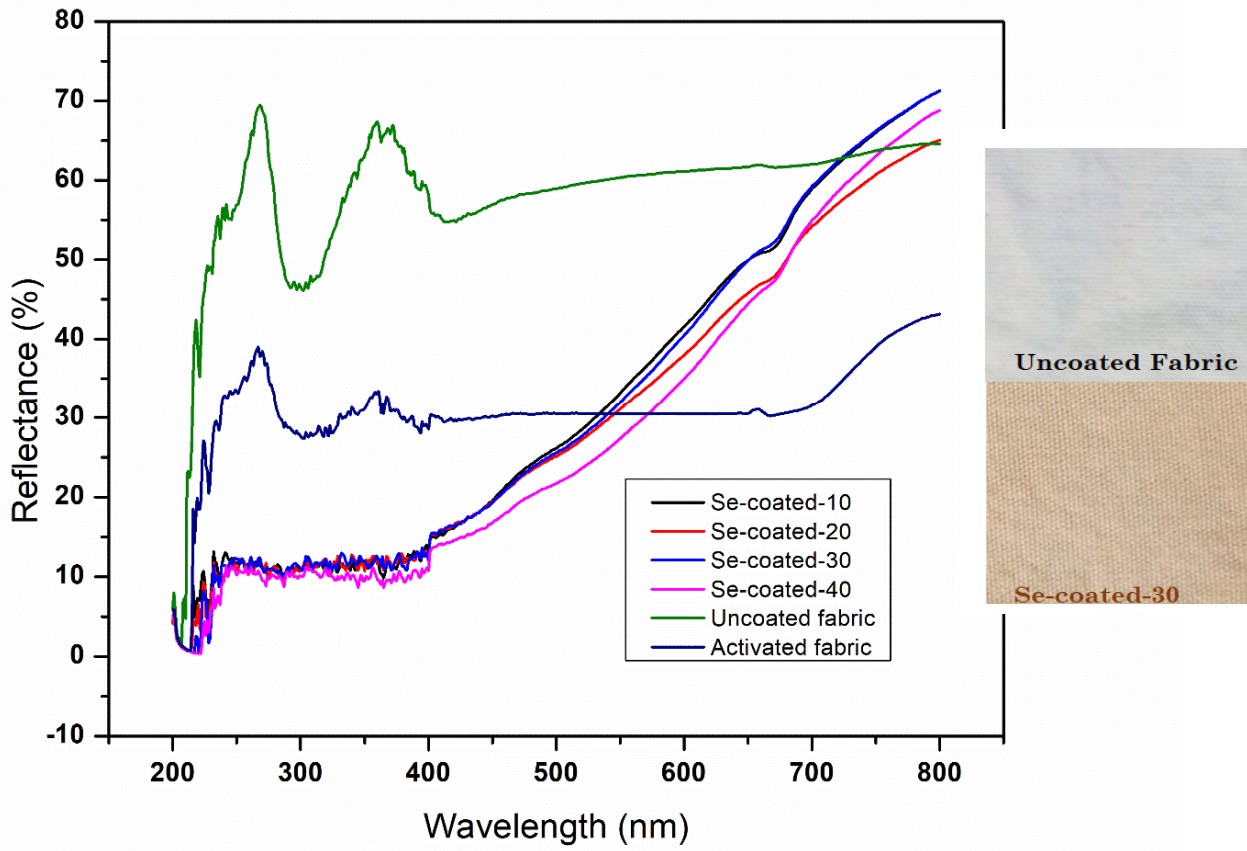

**Figure 2.** UV−Vis reflectance spectra of cotton samples. The images show the visual color change in cotton fabric. The different concentrations of sodium selenite (10–40 mM) were applied on the activated fabric followed by reduction with guava leaf extract and the reflectance spectra were compared in the wavelength range of 200–800 nm. uncoated and activated fabric served as control.

### 3.2.2. Antimicrobial Efficacy of Coated Samples

The coated samples (Se-coated-10, Se-coated-20, Se-coated-30, and Se-coated-40) were screened for their antimicrobial activity against *E. coli*, *S. aureus*, *K. pneumonia*, and *C. albicans* (Table 1). It was observed that 30 mM sodium selenite concentration showed a maximum zone of inhibition for all the microbes selected; a further increase in salt concentration did not have any effect on antimicrobial activity (Figure 3). Hence the Se-coated-30 was chosen for further experiments.

**Table 1.** Shows ZOI (mm) around Se-coated samples against different microbial strains on agar plate. Data are represented as mean with standard deviation (*n* = 3).

| Zone of Inhibition (mm) | | | | |
|---|---|---|---|---|
| Samples | *E. coli* | *S. aureus* | *K. pneumonia* | *C. albicans* |
| Se-coated-10 | - | 10 ± 0.3 | 12 ± 0.2 | 12 ± 0.4 |
| Se-coated-20 | 12 ± 0.5 | 20 ± 0.5 | 23 ± 0.3 | 20 ± 0.3 |
| Se-coated-30 | **16** ± 0.3 | **32** ± 0.7 | **26** ± 0.3 | **35** ± 0.4 |
| Se-coated-40 | 16 ± 0.5 | 32 ± 0.4 | 28 ± 0.1 | 36 ± 0.3 |

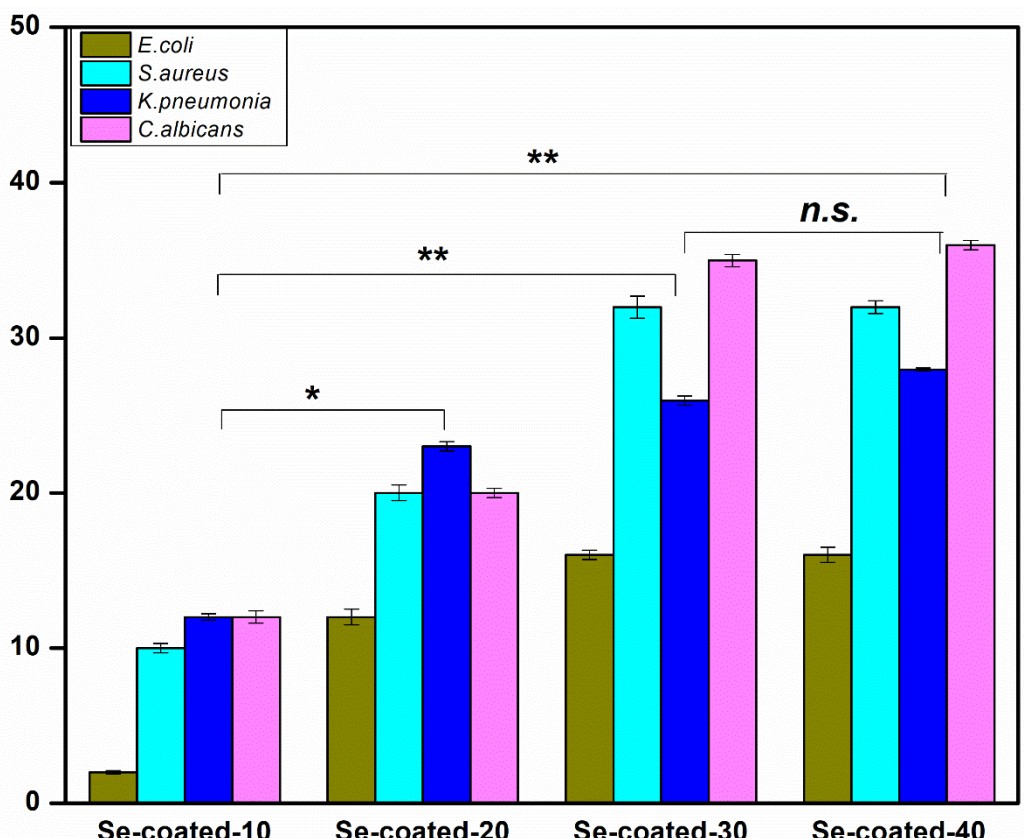

**Figure 3.** Antimicrobial activity of Se-coated samples against three bacterial and one fungal strain. Data are represented as mean with standard deviation ($n = 3$). * $p \leq 0.05$, ** $p \leq 0.01$ represent statistical significance between Se-coated samples (Mann–Whitney test) and n.s. = no significance.

Antibacterial Activity of Se-Coated-30

Previous studies have shown good antibacterial activity of Se NPs in colloidal solutions, indicating their potential to be used in biomedical applications [5,39]. In the present study, the antibacterial effect of the Se-coated-30 was evaluated in terms of its ZOI (Figure 4). The efficacy of Se-coated-30 against the selected Gram-positive and Gram-negative strains is in the following order: *S. aureus* (32 mm) > *K. pneumonia* (26 mm) > *E. coli* (16 mm) in terms of ZOI around the fabric (Table 1). The mode of action of Se NPs against bacteria is multiple, which is beneficial for controlling the development of resistance in microbes. In a previous study, the mechanism of action of guava extract mediated colloidal Se NPs was studied against *E. coli* and *S. aureus*. It was reported that these NPs disrupt the bacterial cell wall and cause nuclear damage [5]. Huang et al. 2019, who investigated the mode of action of Se NPs on bacterial cells, reported internal ATP depletion, ROS production, and altered membrane potential as the reasons for antibacterial activity [40]. Hammad et al. 2020, synthesized microbe-assisted Se NPs; they reported that Se NP-treated *E. coli*, *S. aureus*, *K. pneumonia* and *P. aeruginosa* had increased levels of ROS [39].

Antifungal Activity of Se-Coated-30

Similarly, antifungal activity was evaluated against *C. albicans*, *C. glabrata* and *C. tropicalis* as shown in Figure 5. Effectiveness was evaluated in terms of ZOI. The results indicated that no fungal growth was seen around the Se-coated-30, and a clear ZOI was observed against the selected fungal strains. The coated fabric was the most effective against *C. glabrata* with a ZOI of 45 mm; *C. albicans* and *C. tropicalis* gave a ZOI of 35 mm each (Table 1). A similar ZOI in these two *Candida* species may be due to a similar mechanism of action of Se NPs on these strains. To ascertain that the above results for both bacterial and fungal strains were solely due to Se NP deposition on fabric surfaces, the following controls were also

included in the study: KOH (3 M), salt (30 mM), or guava extract (10%) coated fabric. The Se-coated-30 showed a comparatively bigger ZOI against fungus than the bacterial strains and hence can be used as healthcare textiles to reduce infection and contamination.

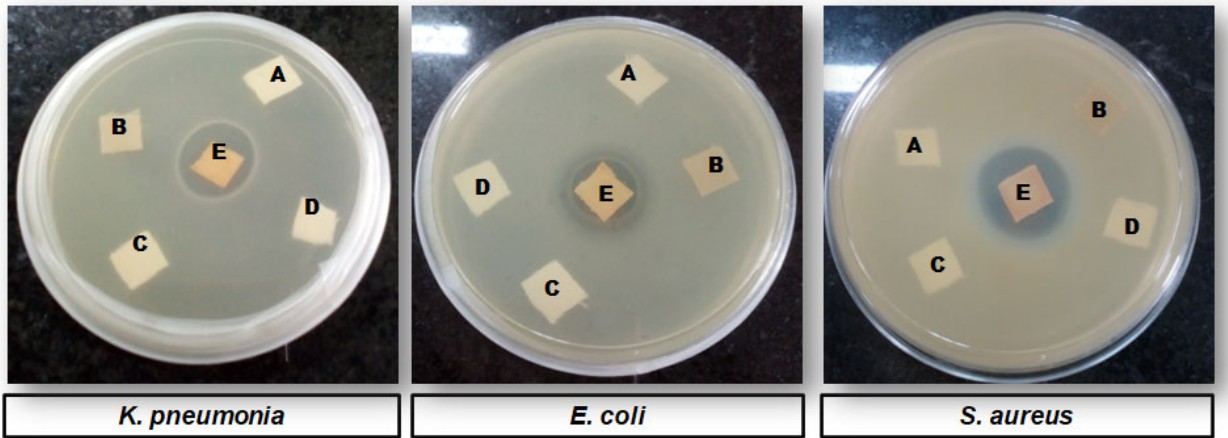

**Figure 4.** Agar diffusion assay of (1) *K. pneumonia* (2) *E. coli* and (3) *S. aureus* showing ZOI around the Se-coated-30 fabric E where A, B, C and D represent controls: sodium selenite salt, guava extract, uncoated and activated fabric samples, respectively.

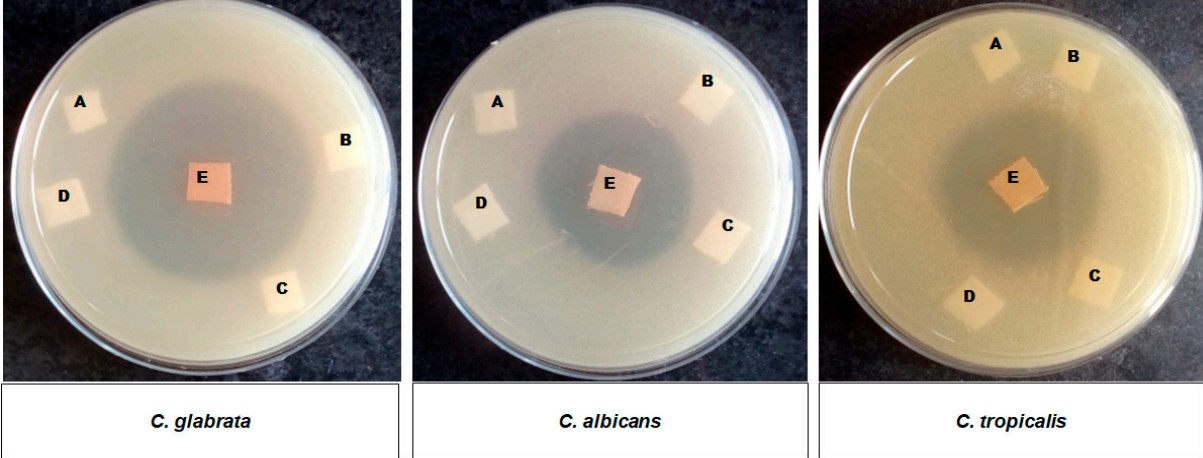

**Figure 5.** Agar diffusion assay of (1) *C. glabrata* (2) *C. albicans* and (3) *C. tropicalis* showing ZOI around the Se-coated-30 fabric E, where A, B, C and D represent controls: sodium selenite salt, guava extract, uncoated fabric and activated fabric samples, respectively.

### 3.2.3. FT-IR Spectra of Se-Coated Fabric

Figure 6 shows the Fourier transform infrared spectroscopy (FT-IR) spectra of uncoated fabrics, activated fabrics, and Se-coated-30. The intensity of the OH peak (3330 cm$^{-1}$) after KOH treatment/activation increases. This may be due to the increase in free primary OH groups on cellulosic cotton, which are formed due to the breaking of intra- and intermolecular H-bonds between the cellulosic fibers. The results are in agreement with previous papers [30]. The appearance of a new peak at 788 cm$^{-1}$ in the Se-coated-30, which is absent in all the other fabric samples, is attributed to Se-O [41]. Table 2 represents the characteristic peaks obtained in the FT-IR spectra of all samples.

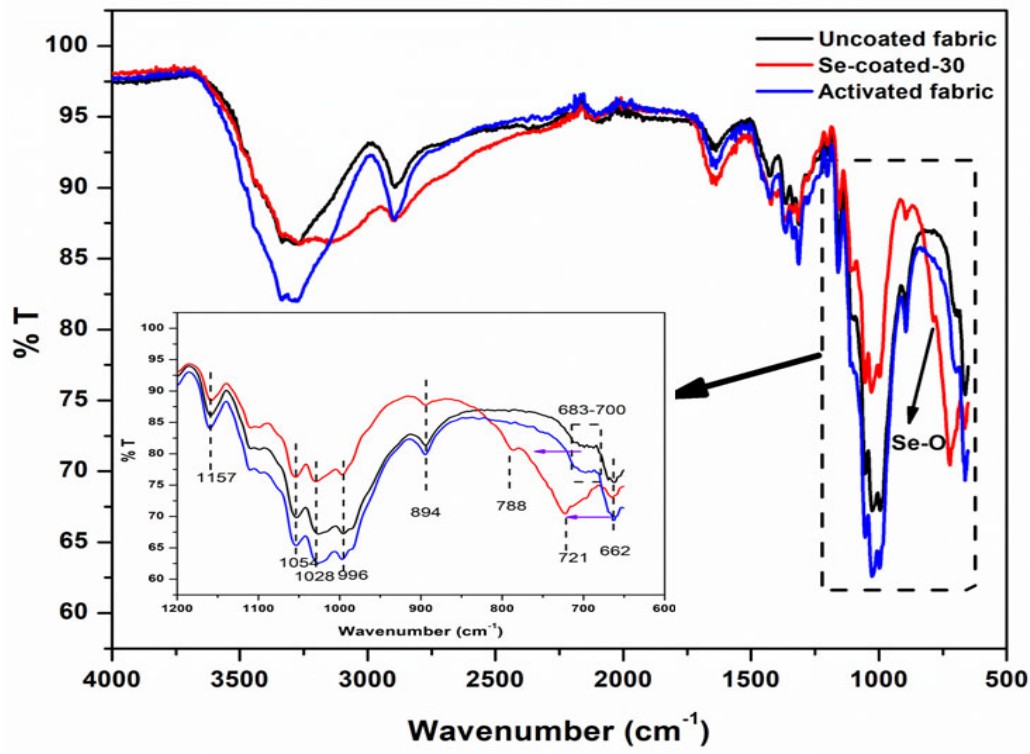

**Figure 6.** FTIR of Uncoated, Activated, and Se-coated fabric samples. Inset shows region between $(600-1200)$ cm$^{-1}$.

**Table 2.** Showing frequencies (cm$^{-1}$) of Uncoated, Activated and Se-coated-30 fabric samples in FTIR.

| Experimental Peaks Frequency (cm$^{-1}$) Obtained | | | | | |
|---|---|---|---|---|---|
| Literature (cm$^{-1}$) | Uncoated Fabric | Activated Fabric | Se-Coated-30 | Peak Characteristics | Ref. |
| 3570–3200 | 3330 | 3330 | 3316–3083 | O6-H str | [42] |
| 3000–2800 | 2886 | 2892 | 2892 | C6-H str | [42] |
| 1430 | 1424 | 1423 | 1420 | CH wagging | [42] |
| 1372 | 1365 | 1363 | 1366 | CH bending | [42] |
| 1163 | 1156 | 1160 | 1157 | C5-O-C1 | [43] |
| 893 | 894 | 894 | 894 | C1-O-C4; β-glucosidic bond | [43] |
| 760–870 | - | - | 788 | Se-O sym str | [41] |

### 3.2.4. XRD

To investigate the changes in crystal structure of raw cotton cellulose after alkali activation and by the incorporation of Se NPs, X-ray diffraction was carried out in the 2θ range of 10–80°. Figure 7 illustrates the XRD patterns of uncoated, alkali-treated, and Se NP-modified cotton fabric. The strong peaks exhibited in uncoated cotton fabric at 14.5, 16.03, 20.07, 22.8, and 34.7 are characteristic peaks of (101), (101), (012), (002), and (040) reflections of cellulose I in crystalline form, respectively [44]. After alkali treatment, it was observed that diffraction peak (002) became sharper with the KOH activation. There is one more peak that appeared at around 25°, which could be due to the changes of cotton fabric's native cellulose I to cellulose II after activation with high KOH concentration [45]. Figure 5 shows the XRD pattern of Se NP-coated fabric, with diffraction peaks other than those attributed to cellulose at 2θ values of 22.24° (112), 23.7° (022), 32.05° (051), 36.20° (441) and 37.20° (351) being attributed to the crystallographic planes of selenium, matched with JCPDS card no. 76–1865 [46,47]. By employing Scherrer's equation, the average crystallite size was estimated as 17 nm. Therefore, the results showed the successful deposition of in situ synthesized Se NPs on the cotton fabric.

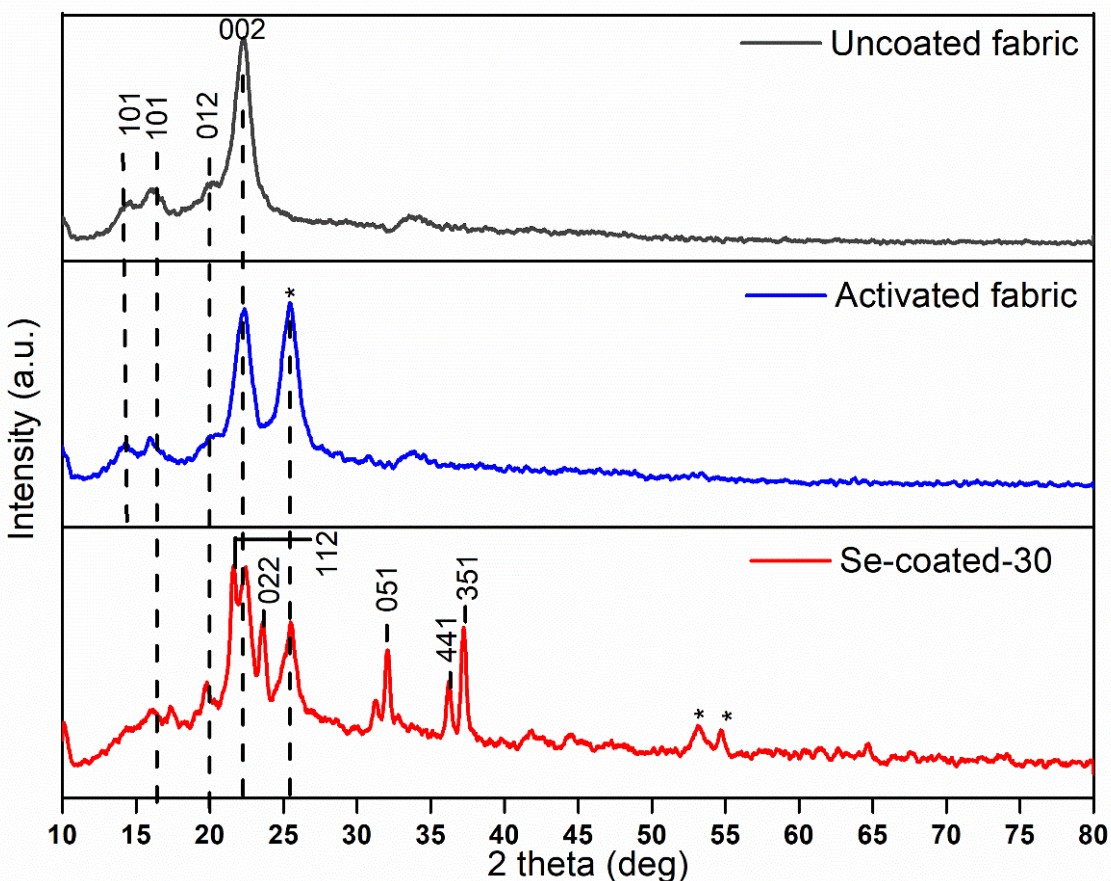

**Figure 7.** XRD pattern of Uncoated, Activated, and Se-coated-30 fabric samples in the 2θ range of 10–80° were compared as shown.

3.2.5. SEM

Figure 8i,ii display the SEM images of uncoated fabric and Se-coated-30. The SEM images of uncoated fabric indicate the clean and smooth longitudinal fibril structure of the fibers without any contaminating particles on their surfaces. Figure 8ii shows that Se NPs with high surface coverage due to their fine, flake-like morphology were formed on Se-coated-30 fabric. In addition, the image shows a rough surface with hierarchical flake structures formed by the in situ deposition of Se NPs on cotton fabric.

The surface chemical elements were analyzed by EDX analysis. Figure 8iii,iv demonstrate the EDX spectra of uncoated fabric and Se-coated-30 with their elemental compositions. The significant peaks of C and O were detected on the uncoated surface, as shown in Figure 8iii. A peak at 2 keV corresponds to Au, which is due to the gold coating of fabrics for SEM imaging. Figure 8iv represents the EDS spectra of Se-coated-30, which shows two new peaks at 1.45 keV and 11.2 keV that were attributed to Se. The weak signals for the presence of Na and K on the coated surface were due to the sodium salt of selenium and unreacted KOH used for cotton activation, respectively. Due to the small size of NPs, a relatively lesser amount of selenium was detected on the surface of the Se-coated-30 sample (8%). Hence, the combined result of SEM and EDS is strong evidence of the formation of Se NPs on the fabric's surface.

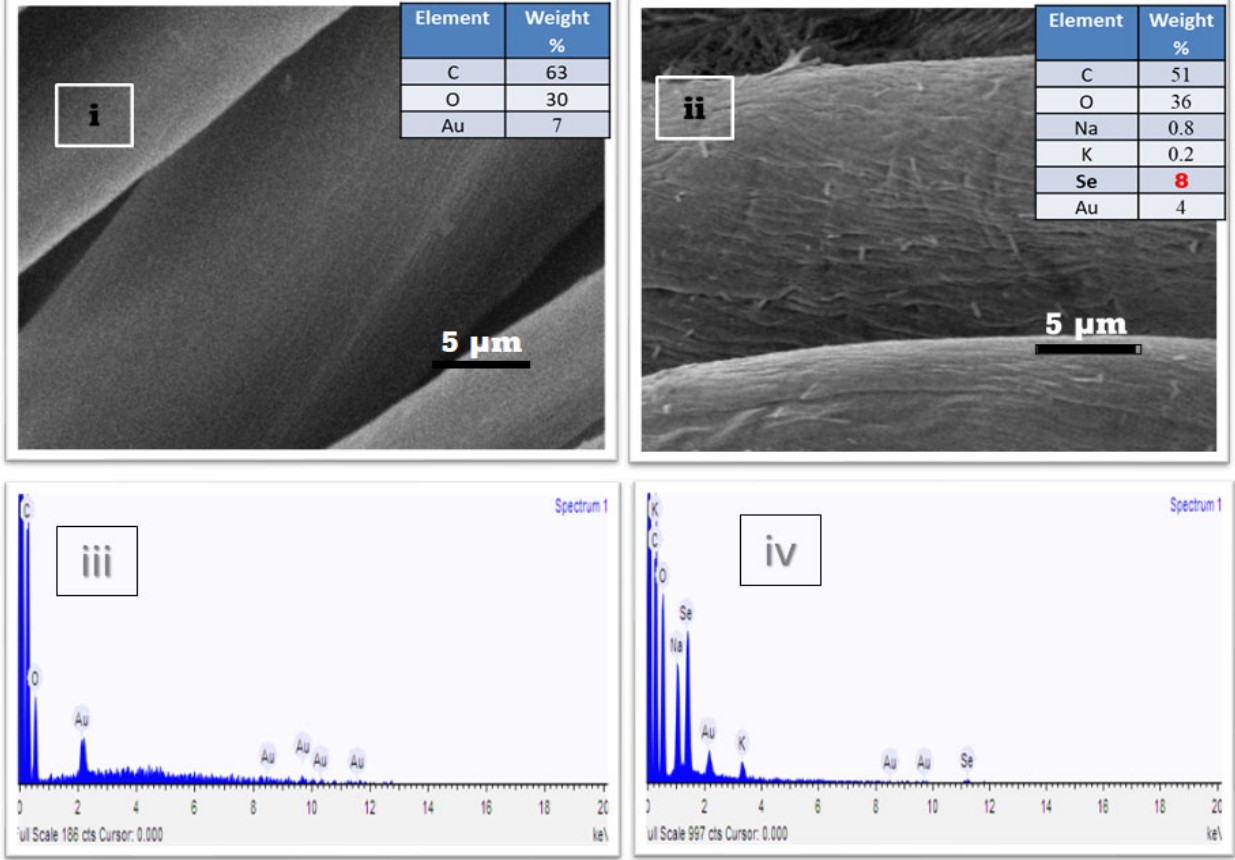

**Figure 8.** Scanning Electron Microscope and EDX spectra of Uncoated fabric (**i,iii**) and Se-coated-30 (**ii,iv**). The scale bar is 5 μm.

### 3.2.6. ICP-OES

The qualitative and quantitative estimation of Se on the fabric was further done in ICP-OES. The amount of Se NPs loaded on the fabric was found to be 3.65 mg/g of fabric.

### 3.3. Antibiofilm Assay

The antibiofilm potential of Se-coated-30 was tested against *E. coli* (Gram-negative bacteria), *S. aureus* (Gram-positive bacteria), and *C. albicans* (pathogenic yeast). The results showed a significant reduction in biofilm viability on the coated fabric compared to the uncoated fabric, even after 48 h of incubation for all the tested strains. The reduction in attached viable cells on the coated fabric was in the following order: *C. albicans* > *S. aureus* > *E. coli.* The antibiofilm potential of coated fabric against *C. albicans* was interestingly high enough (100% reduction in biofilm) to not allow growth of any viable cells post-incubation compared to the uncoated fabric. A dense layer of biofilm was observed visually on the surface of uncoated fabric and confirmed through colony-forming units ($3.94 \times 10^9$ cfu/mL) on the agar plates. In the case of the bacterial strains *S. aureus* and *E. coli*, the growth reduction of biofilm (R%) on the Se-coated-30 surface was found to be 80% and 60%, respectively. This showed that the coated fabric is more efficient against Gram-positive *S. aureus* than Gram-negative *E. coli.* Additionally, visual evidence also showed biofilm inhibition on the Se-coated-30 surface against the tested strains (Figure S1). The previous work of Wang et al. (2020) demonstrated biofilm inhibition by Au NP-coated fabric after 20 h of incubation [24]. In another study, Se- and Ag-coated fabrics were assessed for antibiofilm activity against *E. coli*, *S. aureus*, and *K. pneumonia* for 24 h [16]. The results showed the potential of Se-coated-30 to be used in medical facilities in face masks, aprons, gloves, filter membranes, bandages, etc., for a longer duration of time without changing or washing (Figure 9).

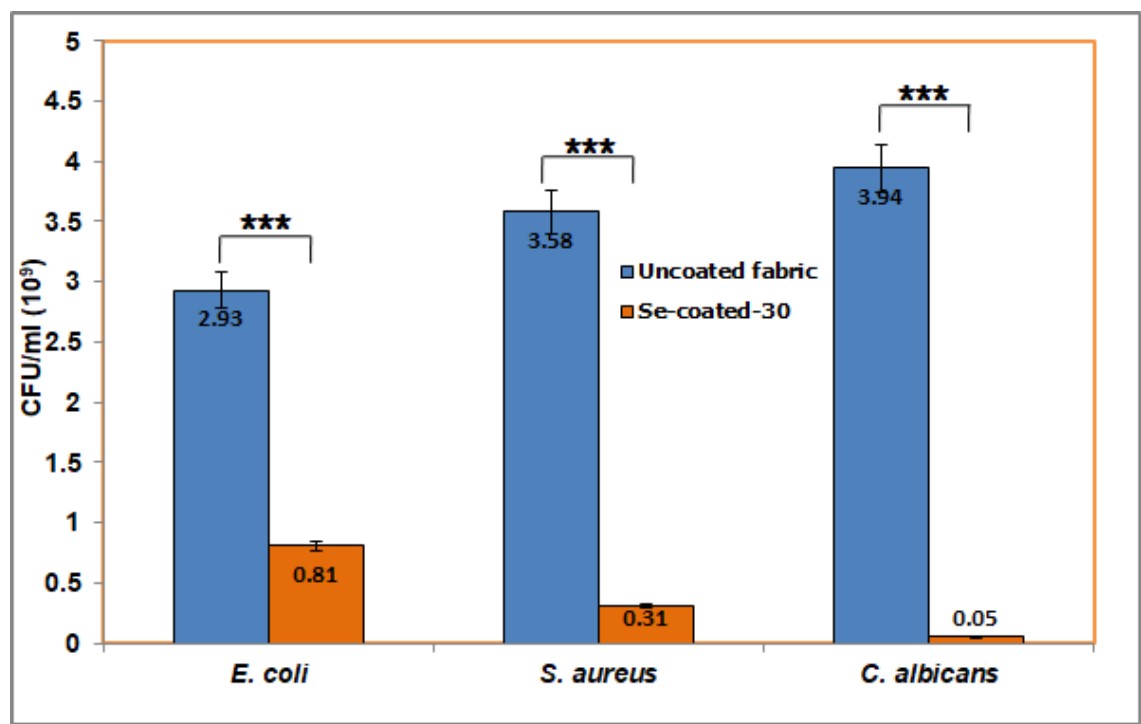

**Figure 9.** Antibiofilm activity of Se-coated-30. Coated fabrics were exposed to *E. coli*, *S. aureus* or *C. albicans* and the biofilm growth was determined using viable counts. Data are represented as mean with standard deviation ($n = 3$). *** $p \leq 0.001$, extremely significant difference as compared to uncoated fabric (Student's *t*-test).

### 3.4. SEM Analysis of Biofilm

SEM analysis for biofilm inhibition was done against *S. aureus*. As seen in Figure 10i, many cells are present together, forming a complex biofilm on the untreated fabric, whereas very few cells are present on the surface of the Se-coated-30 (Figure 10ii). The results show that the Se-coated-30 samples inhibit biofilm formation.

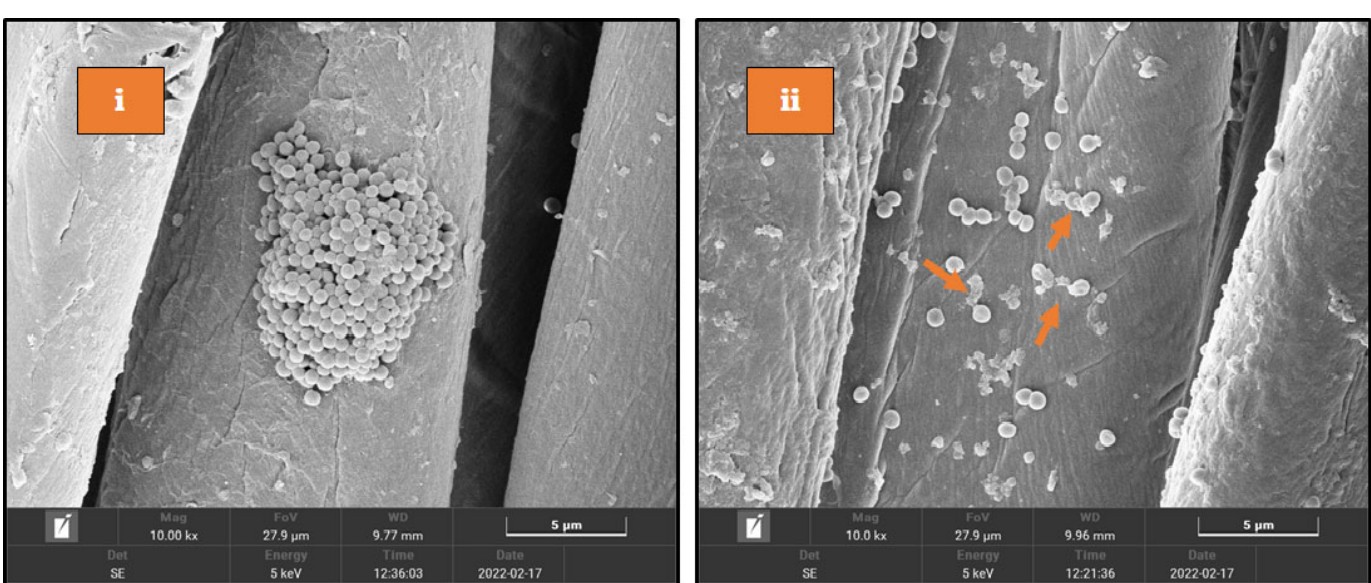

**Figure 10.** SEM images of *S. aureus* cells incubated on the fabric samples for 48 h; (**i**) Uncoated fabric (**ii**) Se-coated-30. The scale bar was 5 μm.

### 3.5. Washing Durability of Se-Coated Fabric

The antibiofilm activity of Se-coated-30 was calculated after 30 washing cycles by the static biofilm assay, as described in the Section 2, and the growth reduction rate (R%) was also calculated against *E. coli*, *S. aureus*, and *C. albicans*. The R% of Se-coated-30 against all the tested strains remained the same even after washing for 30 cycles, as shown in Table 3. The R% for *C. albicans* remains at 99.9% after 30 wash cycles; for *S. aureus* and *E. coli*, the R% decreases negligibly from 80% to 79% and 60% to 58%, respectively, after 30 wash cycles. According to the Student's *t*-test, there was no significant difference between the Se-coated-30 fabric before and after washing. This showed that Se NPs are firmly attached to the fabric, and hence no leaching took place even after 30 wash cycles. A low concentration of Se NPs (3.65 mg/g of fabric) on the fabric showed excellent antimicrobial properties against Gram-positive, Gram-negative, and fungal strains not reported elsewhere. This minimizes the risk of high dosages of these elements and highlights their potential to be used for human welfare.

**Table 3.** Antibiofilm activity of Se-coated-30 after 30 wash cycle. Data are represented as means with standard deviations (*n* = 3). Statistical analysis were performed by the Student's *t*-test.

| Microbes | Percent Reduction (R%) | |
| :---: | :---: | :---: |
| | **Before Wash** | **After 30 Wash** |
| *E. coli* | $60 \pm 2$ | $58 \pm 1$ |
| *S. aureus* | $80 \pm 3$ | $79 \pm 2$ |
| *C. albicans* | $100 \pm 1$ | $99.9 \pm 1$ |

### 4. Conclusions

We report an eco-friendly method to in situ synthesize Se NPs on the surface of fabric. This enhances the antimicrobial property of fabrics against both bacterial and fungal strains. The fabric showed a comparatively large ZOI against the Gram-positive strain *S. aureus* (32 mm) among the tested bacterial strains. In the case of fungal strains, Se-coated-30 showed much higher ZOI comparatively, which was 45 mm against *C. glabrata*. Thus, the Se-coated-30 is more effective against pathogenic fungal strains. The effect of washing on the antibiofilm performance of fabric was evaluated after 30 wash cycles, and it was observed that the fabric had retained excellent antibiofilm performance against *C. albicans*, *S. aureus*, and *E. coli*. The percentage reduction was found to be almost similar before and after 30 wash cycles against *C. albicans* (99%), *S. aureus* (78%), and *E. coli* (58%). The Se amount was found to be 3.65 mg/g of fabric. The fabric's surface was characterized through a surface-limited technique called FT-IR, which easily differentiated the changes taking place in all the fabric samples. XRD was performed to determine the change in crystallinity and nanoparticle formation. Finally, the morphology and elemental composition were analyzed through SEM-EDX. The results indicated that even at a very low concentration of Se, the fabric exhibits strong antimicrobial performance, even after repetitive washing. These fabrics have the potential to be used for mass production in clinical applications such as surgical aprons, face masks, dressing materials, etc.

**Supplementary Materials:** The following supporting information can be downloaded at: https://www.mdpi.com/article/10.3390/fermentation9010018/s1, Figure S1. 24-well plate showing antibiofilm activity of uncoated fabric and Se-coated-30 against microbial strains.

**Author Contributions:** K.M. perform the experiments, design the protocols, and written the manuscript. F.N. and T.A. did the analysis of UV-DRS, XRD and SEM. N.M. provided the facility for the antifungal experiments. M.S. design the experiment, created the idea, analysis of results and edited the manuscript. All authors have read and agreed to the published version of the manuscript.

**Funding:** This research received no external funding.

**Institutional Review Board Statement:** Not applicable.

**Informed Consent Statement:** Not applicable.

**Data Availability Statement:** Not applicable.

**Acknowledgments:** K.M. acknowledges University Grants Commission, Govt. of India, for providing Senior research fellowship. Authors also acknowledges the support provided through the DST PURSE program at CIF, Jamia Millia Islamia.

**Conflicts of Interest:** The authors declare no conflict of interest.

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
