# Peer review of "Development of Cost-Effective, Ecofriendly Selenium Nanoparticle-Functionalized Cotton Fabric for Antimicrobial and Antibiofilm Activity"

_fermentation, doi:10.3390/fermentation9010018_

Round 1
Reviewer 1 Report
The authors have coated cotton fabric with SeNP and characterized the material using different techniques. I recommend the article to be published as is.
Author Response
Thanks a lot for your esteemed recommendation.
Reviewer 2 Report
The article " Development of Cost effective, ecofriendly Selenium Nanoparticles functionalized cotton fabric for antimicrobial and antibiofilm activity" describes the use of Se NPs as excellent antibacterial, antifungal and antibiofilm agent. In order to increase the usefulness and significance of the study, it needs a major revision before being considered suitable for readers and there are some points to overcome for acceptance.
In this article, author focused on Development of Cost effective, ecofriendly Selenium Nanoparticles for antimicrobial and antibiofilm activity. However, the introduction of these sections is quite plain, not in deep. It is recommended to tone up the introduction section.
Please provide SEM images of S. aureus cells incubated on the fabric samples for 48h with original scale bar.
Please provide at least 1 image to see the inhibition. Crystal violet is a gold standard but an initial step method. Or at least show a microtiter plate with biofilms with inhibition and add in supplementary file.
Author represented antibiofilm efficacy of Selenium Nanoparticles by determining CFU count only. It is recommended to check antibiofilm activity using confocal laser scanning microscopy, or Fluorescence microscopy.
Statistical analysis of Antibiofilm activity of Se-coated-30 after 30 wash cycle in Table 3 is missing.
Statistical analysis was performed using a mean value of data are represented and standard deviation. I would also suggest plotting individual data points on graphs instead of/alongside the means as this increases transparency and gives the reader a clearer idea of whether the assumptions made by statistical analysis. Additionally, you could use a Kruskal-Wallis test followed by Mann-Whitney tests adjusted for multiple comparisons to compare with untreated control.
It is suggested a moderate English revision by an English native speaker in order to polish text from typos and imperfections.
Line 387 Author discussed about mechanism of action of Se NPs. Provide valid reference or data for the same or remove the sentence.
Use consistent style for degree Celsius sign: either 37°C or 37 °C. Need to be check and correct carefully throughout the manuscript.
Unwanted spacing and typo mistakes throughout the manuscript. Need to be check and correct carefully.
Double check the way of adding references in the main text body and reference section as per journal guidelines.
Reviewer 3 Report
Please see my comments in the attached file.

Round 2
Reviewer 2 Report
Author addressed all comments carefully so I endorse this article for the publication.
Author Response
Thank you for your useful comments and suggestion